

# Accounting for spatiotemporally correlated errors in wind speed for remote surveys of methane emissions

Bradley M. Conrad & Matthew R. Johnson

Energy and Emissions Research Laboratory, Department of Mechanical and Aerospace Engineering, Carleton University, Ottawa, ON K1S 5B6, Canada

*Correspondence to*: Matthew R. Johnson (matthew.johnson@carleton.ca)

**Abstract**

Spatiotemporally correlated errors in wind speeds estimated via numerical weather prediction (NWP) models – where estimated wind speeds and associated uncertainties at similar times or at nearby sites are likely to be correlated – are an important, but to date neglected, determinant of overall uncertainties in remote surveys of emissions in the oil and gas and other sectors. In this work, we develop a methodology to model such errors using publicly available, anemometer-measured wind speeds at weather stations within a region of interest. The method is parsed into two: (1) creation of a region-averaged wind speed error model to provide the probability of the true wind speed given an NWP model-estimated wind speed while ignoring autocorrelation, and (2) development of a Gaussian copula model for the region of interest (ROI) to capture spatiotemporal autocorrelation. The combined model for total wind uncertainty, including spatiotemporally autocorrelated errors, is demonstrated using the oil and gas-producing region of northeastern British Columbia, Canada as a case study. We also provide additional combined models for the Canadian provinces of Alberta and Saskatchewan, the U.S. state of North Dakota, and Colombia. Finally, we share a simple python code to interface with these models and to simplify application by others. The combined models show varying correlations in wind speed errors, which are attributable to the variability of terrain in the ROIs and the relative accuracy of different NWP models. Results further reveal how temporal correlations and hence uncertainties in aggregated emissions can be minimized through remote survey design, where waiting at least two days before revisiting a site and phase shifting re-surveys by approximately six hours can avoid both near-field and diurnal patterns in temporal autocorrelation.

## 1 Introduction

Accurate quantification of methane emissions is critical for directing mitigation, tracking reductions, and informing climate policy. Remote sensing technologies, especially aerial and satellite-based platforms, have emerged as powerful tools for detecting and estimating methane sources. These technologies support development of regional, sector-specific source distributions (Cusworth et al., 2022; Kunkel et al., 2023; Scarpelli et al., 2024; Sherwin et al., 2024; Yu et al., 2022), as well as both measurement-informed (Fosdick et al., 2025; Santos et al., 2024) and measurement-based inventories (Conrad et al.,





2023a, c; Johnson et al., 2023). However, the accuracy of remotely sensed methane emission estimates is highly sensitive to atmospheric transport processes, particularly wind speed and direction, which govern the dispersion of methane plumes.

A common challenge for these approaches is the need to estimate a representative wind speed to convert observed methane enhancements into emission rates or fluxes. In the absence of in situ wind speed measurements, wind speeds are

commonly estimated from public or commercial numerical weather prediction (NWP) models, such as the High-Resolution Rapid Refresh model (HRRR) and the 12-km analysis from the North American Mesoscale Forecast System (NAM12). These estimates may be spatially interpolated, temporally averaged, and vertically adjusted using assumed wind profiles to match the source height for emission calculations (Johnson et al., 2021; Thorpe et al., 2024). For example, satellite measurements using the Integrated Mass Enhancement or Cross-Sectional Flux methods rely on an "effective wind speed,"

which is typically the temporally averaged 10-m wind speed from an NWP model that is further scaled through a calibration function (e.g., Varon et al., 2018). Aerial measurements may follow similar approaches (Branson et al., 2021; Thorpe et al., 2021) or in the case of LiDAR sensors may use measured plume heights and an assumed boundary layer profile to adjust 10-m wind speed estimates from NWP models to obtain wind speed estimates at a characteristic plume height (e.g., Thorpe and Kreitinger, 2024). Regardless of the measurement platform, the performance of NWP models in capturing local wind

conditions is a key source of uncertainty in remotely derived methane emission estimates that to date has been difficult to quantify.

In the context of aerial or satellite surveys over multiple sources or sites, there is an added challenge whereby wind speeds (and hence their uncertainties) at nearby locations are likely to be correlated. Neglecting this autocorrelation when aggregating sources to produce an inventory will artificially reduce the contribution of wind speed precision error to that of

the overall inventory. More generally, uncertainties in wind speed are expected to be correlated in both space and time, where errors at adjacent locations and similar times would be related, but errors at two far away locations or similar locations at two very different times are more likely to be independent. Region-specific wind speed error models that consider error autocorrelation are thus essential for accurate methane inventories with robustly characterized uncertainties. Unfortunately, such models are conspicuously absent in the literature.

In this manuscript, we detail a methodology to probabilistically model the true wind speed at an arbitrary location and time from gridded, discrete-time NWP model estimates. The developed model explicitly considers spatiotemporal autocorrelation of wind speeds and associated correlation of uncertainties in an easy-to-implement framework. Using the oil and gas-production region of northeastern British Columbia, Canada as a case study, we then use the presented methodology to develop a comprehensive wind speed error model and subsequently show how this model influences a provincial methane

emissions inventory derived from aerial survey data relative to a simpler model (Conrad et al., 2023b) used in our previous work (Conrad et al., 2023a, c; Johnson et al., 2023). Next, we analyze different time periods of case study data to investigate annual variability of wind speed errors and demonstrate the consistency of the modelling method. By creating additional





models for the U.S. State of North Dakota, we then illustrate how our modelling approach can be used to compare regional performance of different NWP models, providing a means to choose the best wind speed data for a given region of interest to minimize uncertainties in methane inventories derived from large scale remote surveys. Finally, referring to Appendix A, we derive six additional models for the remaining oil and gas-producing provinces of Canada (Alberta and Saskatchewan), the

U.S. state of North Dakota, and Colombia, and share a simple python code (see Code and data availability) to describe, visualize, and interact with the created models for the purposes of Monte Carlo simulations of wind speed error.

## 2    Methods

### 2.1    Case study

#### 2.1.1    Region of interest

We present our wind speed error modelling approach using a case study of the oil and gas-producing region of the Canadian province of British Columbia (BC). BC has been a leader in leveraging sustained provincial-scale measurement campaigns (British Columbia Energy Regulator, 2022) to track emissions, guide regulation, and achieve low methane intensities (e.g., Conrad et al., 2023c; Johnson et al., 2023). Moreover, oil and gas production in BC's Montney and Horn River basins is located immediately downwind of the Rocky Mountains for the prevailing westerlies (Figure 1), with complex topography

and terrain that provides an especially challenging wind modelling example.

The pink polygon in Figure 1 identifies the approximately 122,000 km$^2$ region of interest (ROI) defined for this case study that bounds upstream oil and gas production in the Northeast of the province. The bounds of the ROI were defined as a convex hull around active and shut-in upstream facilities and wells during early 2023 (dilated with a 25 km radius circular kernel in the NAD83 UTM 10N coordinate system) intersected with provincial boundaries (northern and eastern sides). The

ROI contains approximately 1,083 oil and gas facilities and 10,441 wells.



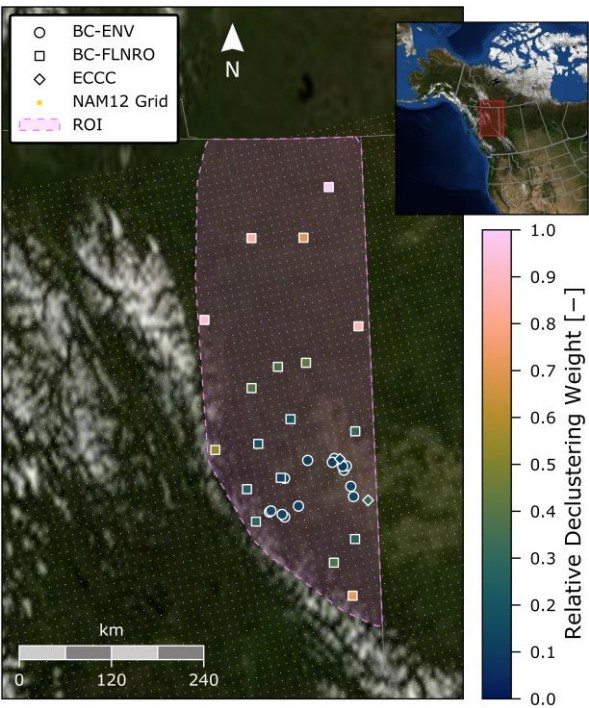

**Figure 1: Case study region of interest (ROI, pink polygon) spanning upstream oil and gas production sites in northeastern BC. Coloured points locate the 35 weather stations that provided independent ground-truth wind speeds for model development, which are coloured according to their relative declustering weights (see Table 1 and Sections 2.2.2 and 2.2.3). The grid at which NAM12 data products are reported underlays the figure in yellow. Underlying surface imagery courtesy of NASA Earth Observatory.**

### 2.1.2 "Ground-truth" measurement data

Anemometer-measured, "ground-truth" wind speeds were acquired from 35 active weather stations within the case study ROI that each report a scalar wind speed and direction on an hourly basis (Figure 1). As summarized in Table 1, these included Environment and Climate Change Canada's (ECCC) surface weather stations (Historical Climate Data, 2025), British Columbia's Ministry of Environment and Parks' Air Quality Network stations (Air Quality and Climate Monitoring: Unverified Hourly Air Quality and Meteorological Data, 2025; Air Quality and Climate Monitoring: Verified Hourly Air Quality and Meteorological Data, 2025), and British Columbia's Ministry of Forests, Lands, and Natural Resource Operations' Wildfire Management Branch's weather stations (BC Wildfire Active Weather Stations and Data, 2025). Two additional ECCC weather stations within the ROI were excluded as they are within 10 km of a station (Fort Nelson Airport; World Meteorological Organization ID 71945) that inputs data to the NAM12 NWP model used for the case study demonstration of our methodology (see Section 2.1.3). Station-measured wind data were obtained for May to October (i.e., non-snowy months when aerial surveys can reliably be performed in the ROI) during the calendar years of 2021−2024. Critically, these data are not inputs to the evaluated NWP model and thus constitute an independent "ground-truth" data set in the subsequent analysis.





**Table 1: Ground-truth data sources providing scalar wind speeds for the present case study.**

| Authority/Network | Acronym | Count | Notes |
|---|---|---|---|
| Environment and Climate Change Canada/ Meteorological Service of Canada's Surface Weather Stations | ECCC | 2 | a,b |
| British Columbia Ministry of Environment and Parks – Air Quality Network | BC-ENV | 15 | c |
| British Columba Ministry of Forests, Lands, and Natural Resource Operations – Wildfire Management Branch's Weather Stations | BC-FLNRO | 18 | d |

[a] Excludes 2 weather stations within 10 km of known NAM data source.
[b] The anemometer is *usually* located at 10 m above ground. Measurements represent the one-, two-, or 10-minute period preceding the top of the hour (Technical Documentation: Historical Hourly Climate Station Data, 2025).
[c] Includes verified (quality controlled; to 2022, inclusive) and unverified (not quality controlled, 2023 onward) data.
[d] Anemometer height varies by site, wind speeds were scaled to 10 m using the logarithmic profile in Johnson et al. (2021).

Data available from the individual networks include hourly reported wind speed magnitudes and directions; however, given the potential for poor resolution and high uncertainties in anemometer-measured wind direction, we ignore direction

data and analyze wind speed magnitudes directly. Recommended installation height for anemometers is usually at 10-m above ground level (e.g., Burt, 2012; Manfredi et al., 2008; Oke, 2004), and wind speeds are "most often [averaged over a period of] 10 minutes" (Burt, 2012). However, installation heights and averaging periods for these weather station networks are either not explicitly stated, ambiguous, or non-standard. For example, documentation for the ECCC weather stations (Technical Documentation: Historical Hourly Climate Station Data, 2025) notes that installed anemometers are *usually*

located at 10-m above ground level and report one, two, or ten minute-averaged wind speeds during the period ending at the top of the hour. While the BC-FLNRO network explicitly states a targeted height range for each installed anemometer, there is no available information regarding averaging periods for the BC-ENV and BC-FLNRO networks nor anemometer height for the BC-ENV network. Thus, anemometers for the BC-ENV network are assumed to be at 10-m above ground level and wind speeds reported by the BC-FLNRO network are scaled to 10-m above ground level from the stated anemometer height

using the logarithmic wind profile in Johnson et al. (2021). Finally, throughout this analysis the averaging period is assumed to be 10 minutes for all weather stations, while noting that this will likely provide a model with larger dispersion relative to measurements over longer averaging periods — refer to the Limitations section (Section 4.1) for further discussion.

### 2.1.3   NWP model data (NAM12)

For the case study, we assess the North American Mesoscale Forecast System (NAM; National Centers for Environmental

Prediction (NCEP) et al., 2025) NWP model, which uses the U.S. National Oceanic and Atmospheric Administration's (NOAA) Non-Hydrostatic Multi-scale Model (NMM). This NWP model is appealing for methane inventory development in




Canada as it provides spatiotemporally resolved two-component wind speeds at 10-m above ground level, is freely available from multiple sources online, and provides near-complete coverage of Canada's provinces. Wind speed estimates are available for a forecasted period of 84 h following each data assimilation cycle that occurs every 6 h. These data are provided on NCEP's #218 grid, which has a 12-km resolution at 35°N and is underlaid in yellow in Figure 1. Wind speed

data on this 12-km resolution grid (hereafter called "NAM12" data) were sourced (see Code and data availability section) from the U.S. National Centers for Environmental Information's https server (up to September 15, 2021) and NOAA's Open Data Dissemination program on Amazon Web Services (from September 16, 2021, onwards). Two-component wind speeds were acquired for each hour during the calendar years of 2021−2024 using the latest data assimilation cycle at each time point to minimize forecast lag; data at 00:00, 06:00, 12:00, and 18:00 UTC therefore represent analyses rather than forecasts.

Component wind speeds were summed in quadrature to provide wind speed magnitudes on the 12-km grid for further processing and comparison with ground-truth, weather station-measured, 10-m wind speeds.

### 2.1.4 Data pre-processing

Prior to model fitting, alignment of the measured and NAM12 wind speed datasets was required. Fortunately, all datasets for the case study ROI reported at identical timepoints, the top of each hour. Wind speeds at the specific coordinates of the

weather stations were assigned using a simple nearest-neighbour interpolation of the NAM12 grid points within the NAD83 UTM 10N projected coordinate system. Missing data in the measurement datasets were stricken. Similarly, recognizing that wind speeds over a finite averaging period cannot be exactly zero but may be below the detection limit of an anemometer, we also struck zero-valued wind speeds in the measurement dataset. Following pre-processing, we have $5\,m \times 1$ vectors containing case study data from all of the 35 weather stations (m = 470,014); these are the ground-truth ($\boldsymbol{u}$ [m s$^{-1}$]) and NWP

model-estimated ($\widetilde{\boldsymbol{u}}$ [m s$^{-1}$]) 10-m wind speeds across all weather stations, the corresponding eastings ($\boldsymbol{s}_x$) and northings ($\boldsymbol{s}_y$) of each measurement in projected coordinates [km], and the corresponding time of each measurement in Coordinated Universal Time ($\boldsymbol{t}$ [d]).

## 2.2 Modelling approach

To characterize the true wind speed at the arbitrary locations and measurement times of sources in a remote measurement

survey, we require a probabilistic error model. This model is the probability of the true wind speed ($u$) given the NWP model-estimated wind speed ($\tilde{u}$) at each location (defined by $s_x$ and $s_y$) and time ($t$) of interest for the survey. Critically, this model must consider the spatiotemporal dependence (i.e., autocorrelation) of the true wind speed.

Consider $n$ remote measurement locations/timepoints for which we wish to model the true 10-m wind speed; here, $\boldsymbol{u}$, $\widetilde{\boldsymbol{u}}$, $\boldsymbol{s}_x$, $\boldsymbol{s}_y$, and $\boldsymbol{t}$ are $n \times 1$ real-valued vectors. The spatial vectors may contain duplicate values representing multiple temporal

measurements at one location in space, while the temporal vectors may contain duplicate values representing simultaneous





measurements at different locations. With these definitions, our objective is to derive a probability distribution for the true wind speed with probability density function (PDF)

$$\pi(\boldsymbol{u}|\tilde{\boldsymbol{u}}, \boldsymbol{s}_x, \boldsymbol{s}_y, \boldsymbol{t}) \tag{1}$$

and cumulative distribution function (CDF)

$$H(\boldsymbol{u}|\tilde{\boldsymbol{u}}, \boldsymbol{s}_x, \boldsymbol{s}_y, \boldsymbol{t}) \tag{2}$$

Given known values for $\boldsymbol{s}_x$, $\boldsymbol{s}_y$, and $\boldsymbol{t}$, and estimated wind speeds $\tilde{\boldsymbol{u}}$ from an NWP model, this distribution permits random

sampling of the true wind speed vector $\boldsymbol{u}$, enabling emissions quantification across an entire survey while specifically considering autocorrelation of NWP model error in space and time.

Beyond the potentially high dimensionality of the problem, deriving a model following Eq. (1) is challenging for two reasons. Firstly, the univariate conditional probability density $\pi(u_i|\tilde{u}_i)$ at the $i^{th}$ location and timepoint is a non-negative distribution of sufficient complexity that the *multivariate* density in Eq. (1) is unlikely to be well-represented by standard

analytical multivariate distributions. Secondly, correlation between the $u_i$ is likely to be strongly dependent on the spatial and temporal lags between the locations and timepoints of interest.

### 2.2.1 Copula approach

One logical and easily implemented approach to circumvent these challenges leverages a statistical tool known as a copula. Copulas were first presented by Sklar (1959) and are discussed in abundant detail in the literature (see, for example, Nelsen

(2006)). In this section, we simply present the formal definition and relate it to the present challenge. Sklar's theorem states that the joint CDF of $n$ arbitrary random variates $\left(H(x_1, \dots, x_n)\right)$ can be represented *exactly* as a function of each $x_i$'s univariate marginal CDF $\left(F_i(x_i)\right)$ through an $n$-dimensional function called a "copula" ($C$) (e.g., Nelsen, 2006), that is

$$H(x_1, \dots, x_n) = C\left(F_1(x_1), \dots, F_n(x_n)\right) \tag{3}$$

or in terms of PDFs (e.g., Arrieta-Prieto and Schell, 2022)

$$\pi(x_1, \dots, x_n) = c\left(F_1(x_1), \dots, F_n(x_n)\right) \times \prod_{i=1}^{n} \pi_i(x_i) \tag{4}$$

where $c(\cdot)$ is the copula density. This formulation is useful as it separates independent information about each $x_i$ (defined

by their *marginal* distributions) and the dependence structure of the $x_i$ (defined by the copula). Stated differently, the copula maps the marginal CDFs to the joint CDF (Nelsen, 2003, 2006; Sadegh et al., 2017).

In the present context, introducing our variables of interest, the desired CDF (Eq. (2)) has the form

$$H(\boldsymbol{u}|\tilde{\boldsymbol{u}}, \boldsymbol{s}_x, \boldsymbol{s}_y, \boldsymbol{t}) = C\left(F_1\left(u_1|\tilde{u}_1, s_{x,1}, s_{y,1}, t_1\right), \dots, F_n\left(u_n|\tilde{u}_n, s_{x,n}, s_{y,n}, t_n\right); \boldsymbol{s}_x, \boldsymbol{s}_y, \boldsymbol{t}\right) \tag{5}$$



Here, we have noted that the marginal distributions for the true wind speeds are conditional on the NWP model-estimated wind speeds and locations and timepoints of the remote survey and that the copula (i.e., the dependence structure or autocorrelation) is similarly functional on the locations and timepoints. The key advantage of this approach is that the copula-based model of the desired joint probability can be parsed into two tractable problems. This two-step modelling process requires: (i) a wind speed error model for the marginal distributions at each location and timepoint, $F_i(u_i|\tilde{u}_i, s_{x,i}, s_{y,i}, t_i)$, and (ii) the copula $\left(C(\cdots; s_x, s_y, t)\right)$ that joins them to give the joint CDF. However, we can further simplify the modelling process by deriving a univariate *region-averaged wind speed error model* with CDF $F(u|\tilde{u})$ that is representative of the *average* error in NWP-estimated wind speeds over the ROI (i.e., dropping the location and timepoint dependence of the marginal distributions). Since this error model would be assumed applicable within the ROI regardless of location/timepoint, the marginal distributions ($F_i$) in Eq. (5) are all equivalent ($F_i \equiv F \ \forall \ i \in \{1, \dots, n\}$). This gives the format of the joint CDF that we ultimately seek to model

$$H(u|\tilde{u}, s_x, s_y, t) = C\big(F(u_1|\tilde{u}_1), \dots, F(u_n|\tilde{u}_n); s_x, s_y, t\big) \tag{6}$$

Numerous types of copulas exist for modelling complex dependence structures (see, for example, Nelsen (2006)). In this work, we employ the Gaussian copula, which has the form

$$C\big(F(u_1|\tilde{u}_1), \dots, F(u_n|\tilde{u}_n); s_x, s_y, t\big) = \Phi_n\left(\big[\Phi^{-1}(F(u_1|\tilde{u}_1)), \dots, \Phi^{-1}(F(u_n|\tilde{u}_n))\big]^T; 0, \Sigma(s_x, s_y, t)\right) \tag{7}$$

where $\Phi^{-1}(x)$ is the inverse CDF (quantile function) of the standard univariate normal distribution and $\Phi_n(x; \mu, \Sigma)$ is the joint CDF for the multivariate normal of arbitrary dimension $n$ with $n \times 1$ mean vector ($\mu$, defined here to be the zero vector, $0$) and $n \times n$ covariance matrix ($\Sigma$), which is parameterized by locations and timepoints ($s_x$, $s_y$, and $t$, all $n \times 1$). We choose the Gaussian copula for this work because it reduces the modelling to that of a covariance function or *covariogram* for a Gaussian field, $\Sigma(s_x, s_y, t)$, and algorithms to generate random numbers from a multivariate normal distribution are readily available in mathematical software applications.

### 2.2.2 Region-averaged wind speed error Model

In a previous work, we described a simple method to probabilistically model true wind speeds from NWP model-estimated wind speeds (i.e., a wind speed error model, $\pi(u|\tilde{u})$) based on controlled release experiments assessing a provider of remote methane detection (Conrad et al., 2023b). This model fit the ratio $u/\tilde{u}$, which we denoted as the relative error ratio (RER), to candidate probability distributions such that the wind speed error model had the form

$$\pi_{RER}(u|\tilde{u}) = \frac{\pi_{cand}\left(\frac{u}{\tilde{u}}; \theta\right)}{\tilde{u}} \tag{8}$$

where $\pi_{cand}(x; \theta)$ is a candidate probability distribution with non-negative support. This previous model, which assumes the RER follows the same distribution regardless of the NWP model-estimated wind speed, fit the limited experimental data





well but is overly simplistic here where we have years of hourly wind speed data across 35 unique sites and need not constrain the RER in this way.

In the present study, we use a more-general model for region-averaged wind speed error that uses the true wind speed ($u$) as the random variate (i.e., not the RER) and allows the distribution parameters to be functional on the NWP model-estimated wind speed (i.e., $\boldsymbol{\theta}(\tilde{u})$) such that

$$\pi(u|\tilde{u}) = \pi_{cand}\big(u; \boldsymbol{\theta}(\tilde{u})\big) \tag{9}$$

We considered nine candidate distributions with non-negative support — Burr Type XII, Gamma, Inverse Gaussian, Log-Logistic/Fisk, Lognormal, Nakagami, Rayleigh, Rician, and Weibull — all of which have two parameters except the Rayleigh distribution, which has just one. Moreover, we permitted each parameter $\theta_i(\tilde{u})$ to be a constant ($a$), linear function ($b\tilde{u} + c$), or offset power-law ($d\tilde{u}^e + f$) giving a total of 75 candidate models for $\pi(u|\tilde{u})$. For each candidate model, we calculated a *weighted* maximum likelihood estimate and chose the model that minimized the Akaike Information Criterion (Akaike, 1974) as optimal. Data were weighted to account for biased spatial clustering of weather stations using a custom implementation of Deutsch and Journel's (1997) "DECLUS" cell declustering algorithm. Following recommended practices (e.g., Deutsch, 1989), this algorithm was implemented with a cell size of 147 km, which minimized the declustered mean of ground-truth wind speeds. Declustering weights, normalized to the unit interval, for the 35 weather stations of the BC case study are shown by colour in Figure 1.

### 2.2.3 Spatiotemporal autocorrelation model

To model the covariogram within the ROI for use in the Gaussian copula, we employ geostatistical variography techniques on the $m$ case study data. There is myriad literature regarding the evaluation, characterization, and simulation of autocorrelated geostatistical data – see, for example, Cressie (1993). Here, we outline our approach to modelling $\boldsymbol{\Sigma}(\boldsymbol{s}_x, \boldsymbol{s}_y, \boldsymbol{t})$ following published techniques within this field. We begin by transforming our case study data to follow a normal distribution via a *normal score transform*, which gives an $m \times 1$ vector $\boldsymbol{z}$ corresponding to the random variate in the multivariate CDF in Eq. (7) (here, of length $m$). We employ a transformation that leverages our optimized region-averaged wind speed error model – each element of $\boldsymbol{z}$ is

$$z_i = \Phi^{-1}\big(F(u_i|\tilde{u}_i)\big) \tag{10}$$

and can be interpreted as a function of location and timepoint via $z_i = Z\big(s_{x,i}, s_{y,i}, t_i\big)$. The vector $\boldsymbol{z}$ represents a random sample from a multivariate normal distribution with zero-mean and some covariance structure, the latter of which we can now simulate using the corresponding location and timepoint vectors. However, to avoid bias in the empirical estimation of covariance (e.g., Isaaks and Srivastava, 1988), we characterize the covariance of $\boldsymbol{z}$ by instead modelling its *semivariogram*.

Like covariance, the semivariogram ($\gamma$) is a means to characterize how similar and different $z_i$ are in the near- and far-fields, respectively — i.e., the spatiotemporal autocorrelation of $\boldsymbol{z}$. In the most general sense, the semivariogram is





functional on two specific positions in space and time and is one-half of the variance of the $z_i$ at these positions. Letting the vector $\boldsymbol{r}_i = [s_{x,i}, s_{y,i}, t_i]^T$, the semivariogram is

$$\gamma(\boldsymbol{r}_1, \boldsymbol{r}_2) = \frac{1}{2}\text{var}(Z(\boldsymbol{r}_1) - Z(\boldsymbol{r}_2)) \tag{11}$$

This formulation can be simplified greatly. Referring to Cressie (1993) for further detail, if the random field $Z$ is *intrinsically stationary*, such that it has a constant mean over all values of $\boldsymbol{r}$ and the variance $\text{var}(Z(\boldsymbol{r}_1) - Z(\boldsymbol{r}_2))$ depends only the spacing (lag) between the positions ($\boldsymbol{h} = \boldsymbol{r}_2 - \boldsymbol{r}_1$), then the semivariogram can be restated as

$$\gamma(\boldsymbol{r}_1, \boldsymbol{r}_2) = \gamma(\boldsymbol{h}) = \frac{1}{2}\text{E}\left[(Z(\boldsymbol{r}_1) - Z(\boldsymbol{r}_1 + \boldsymbol{h}))^2\right] \tag{12}$$

where $E[\cdot]$ is the expectation operator. Furthermore, if the random field $Z$ is *second-order stationary* such that the *autocovariance* $\text{cov}(Z(\boldsymbol{r}_1), Z(\boldsymbol{r}_2))$ depends only on the spacing between the positions, then the covariance of two positions separated by $\boldsymbol{h}$ is easily derived from the semivariogram via

$$\text{cov}(\boldsymbol{h}) = \text{cov}(\boldsymbol{0}) - \gamma(\boldsymbol{h}) \tag{13}$$

where $\text{cov}(\boldsymbol{0})$ is the variance of the field, which is equivalent to the *sill* of the semivariogram (see below). Thus, with a semivariogram model, the elements of a covariance matrix can be easily calculated as a function of spatiotemporal lag.

Following Eq. (12), the semivariogram can be empirically estimated from the case study data by

$$\hat{\gamma}(h_x, h_y, h_t) = \frac{1}{2M(h_x, h_y, h_t)}\sum_{i=1}^{M(h_x, h_y, h_t)}\left(Z(s_{x,i}, s_{y,i}, t_i) - Z(s_{x,i} + h_x, s_{y,i} + h_y, t_i + h_t)\right)^2 \tag{14}$$

where positive real-valued scalars $h_x$, $h_y$, $h_t$ are the lags in two-dimensional cartesian space and time and $M(h_x, h_y, h_t)$ is the number of pairs of data in $\boldsymbol{z}$ that are separated by these lags. The (empirical) semivariogram can be simplified by assuming that the semivariogram is isotropic in space. Letting $\boldsymbol{h}_s = h_x\hat{\boldsymbol{x}} + h_y\hat{\boldsymbol{y}}$ be the spatial lag vector and $h_s = \sqrt{h_x^2 + h_y^2}$ be its positive, real-valued magnitude, the empirical semivariogram simplifies to

$$\hat{\gamma}_{st}(h_s, h_t) = \frac{1}{2M(h_s, h_t)}\sum_{i=1}^{M(h_s, h_t)}\left(Z(\boldsymbol{s}_i, t_i) - Z(\boldsymbol{s}_i + \boldsymbol{h}_s, t_i + h_t)\right)^2 \tag{15}$$

where $\boldsymbol{s}_i = s_{x,i}\hat{\boldsymbol{x}} + s_{y,i}\hat{\boldsymbol{y}}$, $\hat{\boldsymbol{x}}$ and $\hat{\boldsymbol{y}}$ are the cartesian unit vectors, and we introduce the subscript "st" to identify that the semivariogram describes space and time. Practically, for a finite dataset, there are limited realizations of the continuous positive variables $h_s$ and $h_t$. Consequently, empirical estimation of semivariograms requires discretization of the data by lags with a defined tolerance – see, for example, Deutsch and Journel (1997) for further discussion.

In theory, one could interpolate the empirical semivariogram to estimate our desired covariance matrix, $\boldsymbol{\Sigma}(\boldsymbol{s}_x, \boldsymbol{s}_y, \boldsymbol{t})$ for arbitrary locations in space and timepoints. However, careful modelling of the semivariogram is required to ensure that it is valid for the simulation of covariance. We must develop a semivariogram model using appropriate functions (or a





superposition of appropriate functions) that are conditionally negative semidefinite, which in turn ensures a positive definite correlation matrix $\Sigma(s_x, s_y, t)$ (e.g., Curriero, 2006; Pyrcz and Deutsch, 2006). For spatiotemporal applications with an isotropic correlation in space, one useful approach to make modelling of the semivariogram more tractable is to separate it into space and time components. We adopt the *product-sum model* of De Iaco et al. (2001) in this work

$$\gamma_{st}^*(h_s, h_t) = \gamma_{st}^*(h_s, 0) + \gamma_{st}^*(0, h_t) - k^* \, \gamma_{st}^*(h_s, 0) \, \gamma_{st}^*(0, h_t) \tag{16}$$

5    where $\gamma_{st}^*(h_s, h_t)$ is the optimal model of the semivariogram, $\gamma_{st}^*(h_s, 0)$ is a marginal spatial semivariogram model at zero temporal lag, and $\gamma_{st}^*(0, h_t)$ is a marginal temporal semivariogram model at zero spatial lag, which are linked by an optimized parameter $k^*$.

We first model the spatial semivariogram. For each available timepoint of the case study data, we calculate an empirical semivariogram for lags at every 25 km, with a bin tolerance of ± 25 km (e.g., the first bin is 0−50 km, the second is 10    25−50 km, and so forth). Rather than taking the arithmetic mean of the data within each bin (as indicated by Eq. (15)), here we calculate a weighted mean for each bin using weights via Richmond's (2002) two-point declustering method. Like the cell declustering described in Section 2.2.2, this method weights the $k^{th}$ data pair based on the inverse of the number of pairs that originate and terminate in the same cell as the $k^{th}$ pair; our implementation of this method uses the same cell size for the earlier cell declustering, 147 km. As in the region-averaged wind speed error model, this space-informed weighting was 15    executed to provide what we believe is a better average empirical semivariogram across the ROI. We then fit candidate semivariogram models to this empirical semivariogram using a weighted least squares algorithm where the weights are proportional to the number of valid data informing each lag bin of the empirical semivariogram.

We consider 21 unique monotonic semivariogram models as candidate fits. Firstly, as a baseline, the constant model which assumes zero spatial correlation. We also consider semivariograms of the form

$$\gamma_{st}^*(h_s, 0) = b_1 + (b_2 - b_1)\gamma_1^0(h_s; b_4) + (b_3 - b_2)\gamma_2^0(h_s; b_5) \tag{17}$$

20    where $\gamma_k^o(h; b)$ is one of four monotonic semivariogram models standardized between zero and one and parameterized by a fitted parameter $b > 0$ (the *range* of the model):

- Exponential: $\gamma_k^o(h; b) = 1 - \exp\left(-\frac{a_{exp}h}{b}\right)$

- Gaussian: $\gamma_k^o(h; b) = 1 - \exp\left(-\left(\frac{a_{gau}h}{b}\right)^2\right)$

- Spherical: $\gamma_k^o(h; b) = \left(\frac{3}{2}\frac{a_{sph}h}{b} - \frac{1}{2}\left(\frac{a_{sph}h}{b}\right)^3\right)\left(1 - H\left(h - \frac{b}{a_{sph}}\right)\right) + H\left(h - \frac{b}{a_{sph}}\right)$

25    - Pentaspherical: $\gamma_k^o(h; b) = \left(\frac{15}{8}\frac{a_{pnt}h}{b} - \frac{5}{4}\left(\frac{a_{pnt}h}{b}\right)^3 + \frac{3}{8}\left(\frac{a_{pnt}h}{b}\right)^5\right)\left(1 - H\left(h - \frac{b}{a_{pnt}}\right)\right) + H\left(h - \frac{b}{a_{pnt}}\right)$

with $H$ being the Heaviside step function and $a_k$ fixed constants such that the value of each $\gamma_k^o(b; b)$ is exactly 0.95. Fitted parameters $b_i$ are constrained such that the overall semivariogram is monotonically increasing ($b_3 > b_2 > b_1 \geq 0$) and the





ranges of the $\gamma_k^o$ are positive and rank-ordered ($b_5 > b_4 > 0$). We consider each of the four models independently (fixing $b_3$ to 0) and all 16 ordered pairs. Finally, we discard any model that is not statistically better than the constant model (via an F-test) and choose the remaining model that maximizes the parameter-adjusted coefficient of determination for the weighted least squares approach.

Next, we model the temporal semivariogram. For each weather station, we calculate the empirical semivariogram for integer-valued hours up to two weeks (336 hours). Due to the large amount of temporal data in the case study, we derive these empirical semivariograms iteratively by pulling a subset of random pairs of data and terminating when more than $10^4$ data pairs with temporal lags less than or equal to two weeks have been obtained. We then estimate an ROI-averaged empirical semivariogram by taking the weighted average across the weather stations for each bin, weighted by the product of
the cell declustering weights shown in Figure 1 and the amount of data at each lag. We continue by fitting candidate semivariogram models using weighted least squares with weights corresponding to the those used in the empirical semivariogram calculation, summed over all weather stations.

In contrast to the spatial semivariogram, the temporal empirical semivariogram has a clear periodic nature that aligns with the 24-h diurnal cycle (see further discussion and results in Section 3.2). Specifically, we observe that wind speed
errors are maximally correlated at lags that are multiples of 12 hours, with peak positive correlations at lags that are multiples of 24 hours. Given the observed diurnal variation, we adopt candidate models that capture this "hole effect" of the empirical temporal semivariogram. These models include monotonic semivariograms ($\gamma_k^0(h_t)$ as in the spatial semivariogram modelling above superimposed with a periodic semivariogram. The candidate periodic semivariograms are the linear combinations of zeroth-order Bessel functions of the first kind $(J_0(x))$ as in Ecker and Gelfand (1997) and Ye et
al. (2015) and a cosine function. For temporal lag $h_t$ in days, the fitted models have the form

$$\gamma_{st}^*(0, h_t) = b_1 + (b_2 - b_1)\gamma_1^o(h_t; b_5) + (b_3 - b_2)\gamma_2^o(h_t; b_6)$$

$$+(b_4 - b_3)\left(1 - \sum_{j=1}^{J} c_j J_0\big(2\pi j h_t + \phi_j\big) - d\cos(2\pi h_t)\right)$$

(18)

Here, we scale the Bessel functions' arguments by $2\pi$ and introduce phase shifts $\phi_j$ to force the $(j+1)^{th}$ peak to occur at a lag of exactly one day; similarly, the cosine's argument has a scaling factor of $2\pi$ to capture the diurnal cycle. Fitted parameters $b_i$ are constrained such that each component semivariogram causes a net nonnegative effect ($b_4 \geq b_3 \geq b_2 \geq b_1 \geq 0$); the ranges of the $\gamma_k^o$ are positive and rank-ordered ($b_6 > b_5 > 0$); the coefficients of the Bessel basis functions are decreasing
$(c_1 > \cdots > c_J)$; the coefficient on the cosine is nonnegative ($d > 0$); and, like $\gamma_k^o$, the periodic component is standardized to start from zero and oscillate around one at large lags. The former requires that





$$\sum_{j=1}^{J} c_j J_0(\phi_j) + d = 1 \tag{19}$$

Two-hundred and eleven candidate semivariogram models are considered with up to ten ($J = 10$) Bessel basis functions: a constant model ($b_2 = b_3 = b_4 = 0$); a constant plus oscillating model (10 total, with $b_2 = b_3 = 0$); one monotonic semivariogram plus oscillating models (40 total, with $b_3 = 0$); and two monotonic semivariograms plus oscillating models (160 total, with all $b_i > 0$). We select the optimal model in the same manner as for the spatial semivariogram.

The final step in characterizing the spatiotemporal semivariogram is to optimize for the constant $k^*$ in Eq. (16). We follow the procedure of De Iaco et al. (2001). First, we calculate a two-dimensional empirical spatiotemporal semivariogram (Eq. (15)) using the same lag bins as the independent spatial and temporal semivariograms. Like the spatial and temporal semivariograms, we use a weighted mean in place of the arithmetic mean for Eq. (15) to obtain an ROI-average semivariogram. As in the temporal semivariogram, the weights here are the product of the declustering weights and count of
data in each lag bin; however, we use the two-dimensional declustering algorithm here for the declustering weights consistent with the spatial semivariogram. We then take the sum of the weights within each lag bin and perform a relative weighted least squares fit to Eq. (16) to optimize for $k^*$.

With a full spatiotemporal semivariogram model, we can now define the desired covariance matrix for arbitrary locations and timepoints. First though, we must calculate the *sill* of the fitted spatiotemporal semivariogram model, which is the value
that a semivariogram approaches or oscillates around as the lag(s) go to infinity. For the spatial semivariogram in Eq. (17), the sill is $b_2$ or $b_3$ depending on the form of the optimized model, while for the temporal semivariogram in Eq. (18), the sill is $b_4$. Denoting the sills of the optimized spatial and temporal semivariograms as $L_s^*$ and $L_t^*$, respectively, De Iaco et al. (2001) show that the sill of the spatiotemporal semivariogram ($L_{st}^*$) is

$$L_{st}^* = L_s^* + L_t^* - k^* L_s^* L_t^* \tag{20}$$

and, under the aforementioned conditions of second-order stationarity, the covariance can be easily computed from the
semivariogram as

$$\text{cov}(h_s, h_t) = L_{st}^* - \gamma_{st}^*(h_s, h_t) \tag{21}$$

Re-introducing the vectors $\boldsymbol{s}_x$, $\boldsymbol{s}_y$, and $\boldsymbol{t}$, the covariance matrix required for the Gaussian copula is therefore

$$\boldsymbol{\Sigma}(\boldsymbol{s}_x, \boldsymbol{s}_y, \boldsymbol{t}) = \begin{cases} \text{cov}(h_{s,ij}, h_{t,ij}) & , \quad i \neq j \\ 1 & , \quad i = j \end{cases} \tag{22}$$

where $h_{s,ij} = \sqrt{(s_{x,i} - s_{x,j})^2 + (s_{y,i} - s_{y,j})^2}$, $h_{t,ij} = |t_i - t_j|$, and the diagonal of the covariance matrix is fixed at unity to ensure that the marginal distributions of each $u_i$ follow the region-averaged wind speed error model.





## 2.3 Model implementation

Recalling Section 2.2, consider again a remote measurement survey with $n$ observations for which we desire the true 10-m wind speed ($n \times 1$ vector $\boldsymbol{u}$). Further assume that emissions have been reported using NAM12 wind speeds ($n \times 1$ vector $\tilde{\boldsymbol{u}}$) and let emission rates scale with some effective wind speed that is a function, $f$, of the 10-m wind speed. We seek true emission rates ($n \times 1$ vector $\boldsymbol{q}$) that update the reported emission rates ($n \times 1$ vector $\tilde{\boldsymbol{q}}$) with bias, uncertainty, and correlation in the wind speeds by

$$q_i = \tilde{q}_i \frac{f(u_i)}{f(\tilde{u}_i)} \tag{23}$$

This analysis could theoretically be executed in a Bayesian framework, but such an analysis would quickly become intractable as the number of observations ($n$), and thus the dimensions of the joint distribution $\pi^*(\boldsymbol{u}|\tilde{\boldsymbol{u}})$, increases. Instead, a Monte Carlo framework can efficiently draw random samples of the vector $\boldsymbol{u}$ and use them in Eq. (23) to update the emission rates and quantify uncertainties.

Steps to generate random samples of the true 10-m wind speed vector to perturb estimated emission rates within a Monte Carlo analysis are as follows (please also see code provided with supplemental information):

1. Calculate the covariance matrix ($\boldsymbol{\Sigma}$) from vectors $\boldsymbol{s}_x$, $\boldsymbol{s}_y$, and $\boldsymbol{t}$ using Eq. (22);
2. For each Monte Carlo iteration, calculate an $n \times 1$ vector of random variates ($\boldsymbol{z}$) from the multivariate normal distribution with zero mean and the calculated correlation matrix;
3. Apply the standard univariate normal cumulative distribution function to each element of $\boldsymbol{z}$ independently, which gives $g_i = \Phi(z_i)$;
4. Invert the normal score transform of Eq. (10) (i.e., "back-transform") to obtain an $n \times 1$ vector of random ground-truth 10-m wind speeds ($\boldsymbol{u}$) with

$$u_i = F^{-1}(g_i|\tilde{u}_i) \tag{24}$$

5. Obtain an $n \times 1$ vector of random updated emission rates ($\boldsymbol{q}$) with

$$q_i = \tilde{q}_i \frac{f(u_i)}{f(\tilde{u}_i)} \tag{25}$$

6. Go to step 2 and repeat for desired number of Monte Carlo iterations.

## 3 Results

### 3.1 Region-averaged wind speed error model for case study

Using the BC case study region of Figure 1 as an illustrative example, Figure 2a compares the measured 10-m wind speeds during May to October of 2021−2024 from the 35 weather stations (i.e., ground-truth data on the vertical axis) versus





NAM12-estimated 10-m wind speeds (horizontal axis), both in units of m s$^{-1}$. The colour scale indicates the mode-normalized, empirical joint density of the underlying data computed via a Gaussian kernel density estimator (KDE; see, for example, Scott (2015) for further details). The difference between the overlaid 1:1 correspondence line and the plotted expected value of the ground-truth wind speed conditional on the NWP model-estimated wind speed (i.e., the empirically

5   estimated conditional mean/expectation, $E_{KDE}[u|\tilde{u}]$) reveals significant bias error in the NWP model's estimates. This is perhaps unsurprising for this region of BC that is heavily forested, immediately downwind of the Rocky Mountains for the prevailing westerlies, and largely in their foothills (complex topography). Spatially coarse, time-averaged/smoothed NWP models can be expected to miss potentially significant microscale variations in wind, which here tend to bias NWP model-estimated wind speeds high.

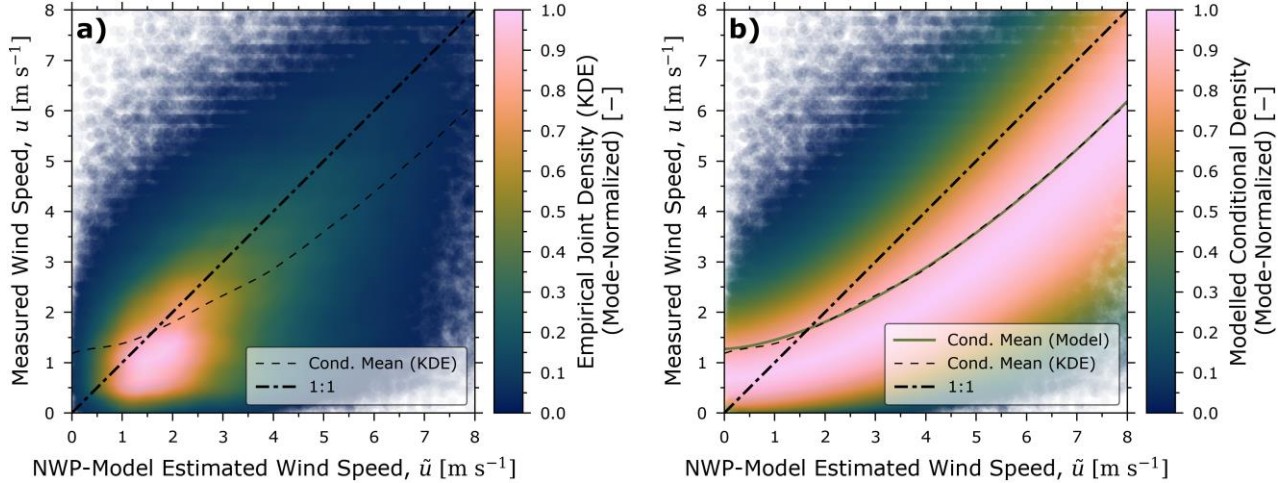

**Figure 2: Summary plot of derived region-averaged wind speed error model for the present case study showing available data points with forecasted and measured wind speeds below 8 m s$^{-1}$. (a) plots the mode-normalized, empirical joint density using Gaussian kernel density estimation (KDE) and the conditional mean $\left(E_{KDE}(u|\tilde{u})\right)$ and (b) plots the modelled conditional density $\pi^*(u|\tilde{u})$ and conditional mean. Good agreement between the central tendencies can be seen in subplot (b) where the modelled and empirical conditional densities closely match.**

A region-averaged wind speed error model was fit to these data following the procedure in Section 2.2.2. The optimal region-averaged wind speed error model followed a Weibull distribution, as is typical for the modelling of site-level wind speed data (e.g., Elliott et al., 2004). The optimized parameters of the distribution both followed the offset power-law candidate such that the PDF and CDF of the optimized model (indicated with a superscript asterisk) are

$$\pi^*(u|\tilde{u}) = \frac{a(\tilde{u})}{b(\tilde{u})}\left(\frac{u}{b(\tilde{u})}\right)^{a(\tilde{u})-1} \exp\left(-\left(\frac{u}{b(\tilde{u})}\right)^{a(\tilde{u})}\right)$$

$$F^*(u|\tilde{u}) = 1 - \exp\left(-\left(\frac{u}{b(\tilde{u})}\right)^{a(\tilde{u})}\right)$$

(26)

20   with shape and scale parameters fitted to (showing coefficients to three decimal points)



$$a(\tilde{u}) = 0.044\,\tilde{u}^{1.551} + 1.504$$
$$b(\tilde{u}) = 0.203\,\tilde{u}^{1.592} + 1.408 \tag{27}$$

Figure 2b plots the same data as Figure 2a but colours the points by their modelled conditional probability $\left(\pi^*(u|\tilde{u})\right)$, which is normalized by the conditional mode for visibility. The conditional mean of the model is plotted alongside the empirically calculated conditional mean, showing near overlapping agreement between the model's and the underlying data's central tendency. Indeed, the modelled and empirical conditional means only deviate notably at forecasted wind speeds exceeding

5   8.5 m s$^{-1}$, where less than 1.2% of the case study data reside.

The goodness-of-fit of the region-averaged wind speed error model is also illustrated in Figure 3, which compares the ground-truth 10-m wind speed for sub-domains of the NWP model-estimated 10-m wind speed. Subplots a−t in the figure plot the empirical CDFs for discrete bins of the forecasted wind in increments of 0.4 m s$^{-1}$ (up to 8 m s$^{-1}$, representing 98.1% of the data) in addition to the modelled CDF $\left(F^*(u|\tilde{u}_{ctr})\right)$ evaluated at the centre of each bin (i.e., $\tilde{u}_{ctr}$). Overlaid in text is

10  the number of valid datapoints within each bin as well as two additional statistics: the root mean squared deviation ($\Delta_{RMS}$) and mean deviation/bias ($\Delta_{bias}$) between the modelled and empirical CDFs. These statistics are calculated using the empirical conditional distribution $\tilde{F}_{KDE}(u|\tilde{u})$ and the region-averaged wind speed error model from Eq. (26). $\Delta_{RMS}$ is equivalent to the root of the Cramer-von-Mises distance

$$\Delta_{RMS} = \left(\int_0^1 \left(F^*(u|\tilde{u}) - \tilde{F}_{KDE}(u|\tilde{u})\right)^2 dF^*(u|\tilde{u})\right)^{\frac{1}{2}} \tag{28}$$

where the integration is performed numerically using the trapezoidal rule. Analogously, the bias is

$$\Delta_{bias} = \int_0^1 \left(F^*(u|\tilde{u}) - \tilde{F}_{KDE}(u|\tilde{u})\right) dF^*(u|\tilde{u}) \tag{29}$$

15  These statistics vary as a function of $\tilde{u}$ and trend only slightly with the quantity of data in each bin. The RMS deviation of the empirical and modelled CDFs is generally less than 0.02 and bias is typically near-zero; indeed, over all of the case study data, $\Delta_{RMS}$ and $\Delta_{bias}$ are approximately 0.017 and 3.5×10$^{-5}$, respectively.







**Figure 3: Assessment of the region-averaged wind speed error model's goodness-of-fit comparing the empirical and modelled cumulative distribution functions (CDFs) for subsets of the training data, parsed into forecasted wind speed bins of 0.4 m s⁻¹ in width. Each subplot shows the number of training data in the bin, the root-mean-squared deviation of the empirical and model CDFs ($\Delta_{RMS}$) and the absolute bias of the model's CDF ($\Delta_{bias}$). A good model fit is observed across all forecasted wind speeds with overall $\Delta_{RMS}$ and $\Delta_{bias}$ of approximately 0.016 and $-3.9\times10^{-5}$, respectively.**

## 3.2 Spatiotemporal autocorrelation of wind speed errors for case study region

With an optimized region-averaged wind speed error model, we proceed by modelling the spatiotemporal autocorrelation within the case study ROI following Section 2.2.3. Figure 4a plots the empirical and modelled spatial semivariograms in lag units of km while Figure 4b plots the temporal semivariograms in lag units of days. To clearly illustrate the sign of spatiotemporal correlation, we plot results both in terms of their semivariogram (left axis) and the their correlogram (right





axis) — i.e., correlation as a function of spatial and/or temporal lag. With the previously noted conditions of second-order stationarity, the spatial correlogram is trivially calculated by

$$\rho_s^* = \frac{L_s^* - \gamma_{st}^*(h_s, 0)}{L_s^*} \tag{30}$$

with analogous equations for the temporal (with subscript t and $\gamma_{st}^*(0, h_t)$) and spatiotemporal (with subscript st and $\gamma_{st}^*(h_s, h_t)$) correlograms.

Data points in Figure 4a illustrate the empirical data at 25-km intervals, coloured by the employed fitting weight (see Section 2.2.3), and the line plots the modelled semivariogram. The best-fitting semivariogram model for the BC case study was the spherical model yielding a positive correlation approaching 0.55 in the near-field and monotonically decreasing towards zero with 95% decorrelation (i.e., the range of $\gamma_{st}^*(h_s, 0)$) occurring at a spatial lag of approximately 95 km. This illustrates how wind errors at locations within ~100 km tend to be correlated. Although, with just 35 data points, the

empirical semivariogram data are quite scattered and the quality of the fit is questionable (one can envision alternative models fitting nearly as well), this is an ROI-specific limitation. For example, the supplemental figure for the additional analysis of North Dakota shows a much-improved goodness-of-fit for the spatial semivariogram attributable to North Dakota's relatively flatter topography.

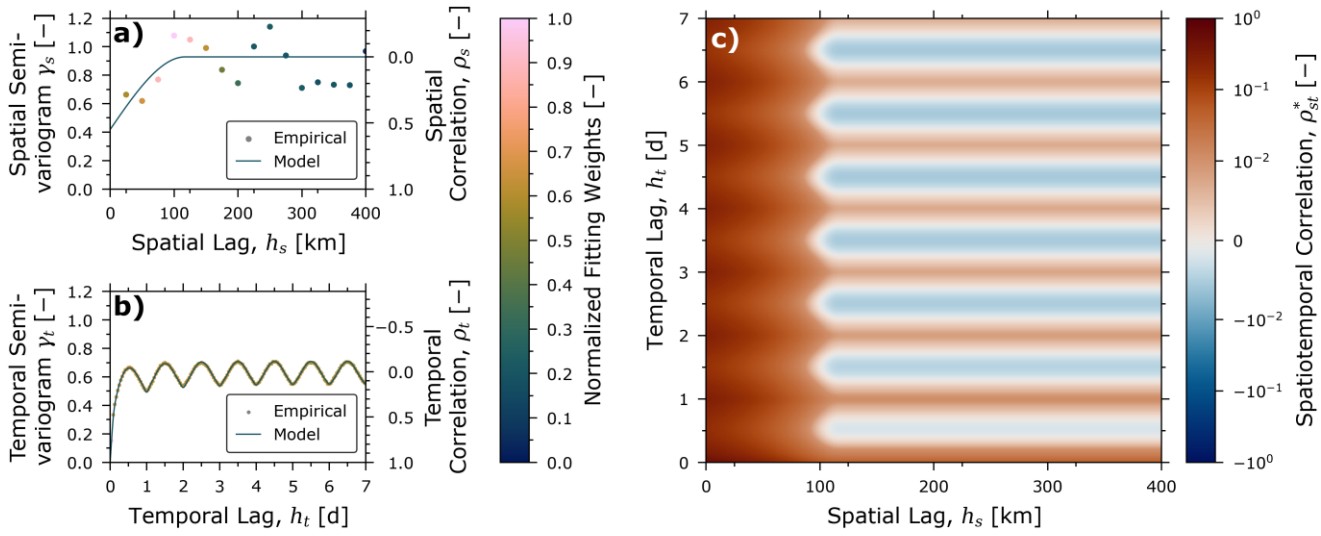

**Figure 4: Summary of empirical and modelled spatiotemporal autocorrelation. (a) and (b) plot the empirical and modelled spatial and temporal semivariograms/correlograms, respectively, and (c) plots the modelled two-dimension correlogram. A poor goodness-of-fit to the spatial semivariogram (a) is expected for this case study due to the low number of unique spatial data (35 weather stations) and corresponding scatter in the empirical semivariogram; see text for additional discussion. The empirical temporal semivariogram (b) is captured remarkably well by the present model with a parameter-adjusted coefficient of variation**

**exceeding 0.9999. The global sill of the spatiotemporal semivariogram is approximately 0.991, close to the ideal value of 1.0 expected following normal score transformation, indicating the quality of the region-averaged wind speed error model.**



The empirical and modelled temporal semivariograms of Figure 4b, almost perfectly overlap (parameter-adjusted coefficient of determination exceeding 0.99996), while revealing high correlation over short intervals and smaller diurnal correlations that persist over longer time intervals. The optimized semivariogram from Eq. (18) combines two exponential semivariograms for $\gamma_1^o$ and $\gamma_2^o$ with ranges of 0.35 and 3.0 days, with an effective range (i.e., 95% decorrelation excluding the oscillating component of the semivariogram) of approximately 1.5 days, and includes seven Bessel basis functions ($J = 7$). The data demonstrate that temporal correlation at small lags can be quite significant, yielding positive correlations greater than 0.63 for lags less than the hourly resolution of the NAM12 data. At large lags, temporal correlations, representing bias over the diurnal cycle, oscillate with an amplitude of approximately 0.13.

Figure 4c shows the two-dimensional spatiotemporal correlogram, created by combining the modelled spatial and temporal semivariograms into a two-dimensional model by optimizing for $k^*$ in Eq. (16). The optimized $k^*$ value of 0.996 corresponds to a global sill for the spatiotemporal semivariogram of 0.976, which is encouragingly close to the idealized global sill of 1.0 expected following a normal score transform. This is a testament to the quality of the fitted region-averaged wind speed error model.

Figures 4a and 4b identify that the modelled sills of the spatial and temporal variograms are non-identical. This suggests an anisotropy in the spatiotemporal data where the overall variance in the ROI is more-strongly affected by variability in space (sill of 0.928) than variability in time (sill of 0.636). However, the nuggets of the semivariograms (i.e., their values at zero-lags) imply that the converse is true for near-field correlation, where we observe stronger correlations in time than we do in space. This has important implications for the timing of remote surveys of methane emissions. The stronger temporal correlation suggest that repeat measurements (i.e., re-visits) should ideally be spaced by more than approximately two days (the effective range of the temporal semivariogram) to reduce correlated errors in inferred emissions. Moreover, due to the diurnal variation in temporal correlations, phase shifting repeat measurements by approximately 6 hours (a quarter day) could theoretically minimize the correlated errors in inferred emission rates. Of course, this is not possible for observations by sun-synchronous satellites but could be accommodated by careful planning of aerial surveys.

### 3.3 Effect of correlated wind speed errors on basin-scale methane inventory

To illustrate the effect of correlated wind speed errors on a jurisdiction-level methane inventory, we recalculate the inventory of Johnson et al. (2023) for BC in 2021. In this analysis, to demonstrate how in the limit of infinite measurement data *uncorrelated* wind speed errors cause uncertainties to diminish towards zero, we replicate the aerial survey data 2, 5, 10, 50, and 100 times with each replication being shifted in time to simulate additional measurements on additional days. We then compute the measured source portion of the inventory three different ways: first, using the RER scaling model of Conrad et al. (2023b) applied to the present training data; second, using the present region-averaged wind speed model (i.e., *without* autocorrelation), and third, using the present *combined* model (i.e., *with* autocorrelation). Referring to Johnson et al. (2023)



for further details, the provincial inventory for BC in 2021 is based on 527 quantified sources at 508 sites comprising 601 active facilities and 904 active wells, which are perturbed within a Monte Carlo framework and scaled to the population of facilities/wells in the province. For the present comparison, minor methodological differences to simplify and accelerate this analysis are employed including: 1) all flight passes with unquantified detections are discarded, 2) missed passes are not

perturbed using the Bayesian analyses described by Johnson et al. (2023), and 3) sample size uncertainty quantified via bootstrap is ignored. For these reasons, the absolute magnitude of the calculated inventory is not directly comparable to Johnson et al.'s (2023) result and results are instead normalized by the recalculated measured source inventory using the combined model. Results are summarized in Figure 5.

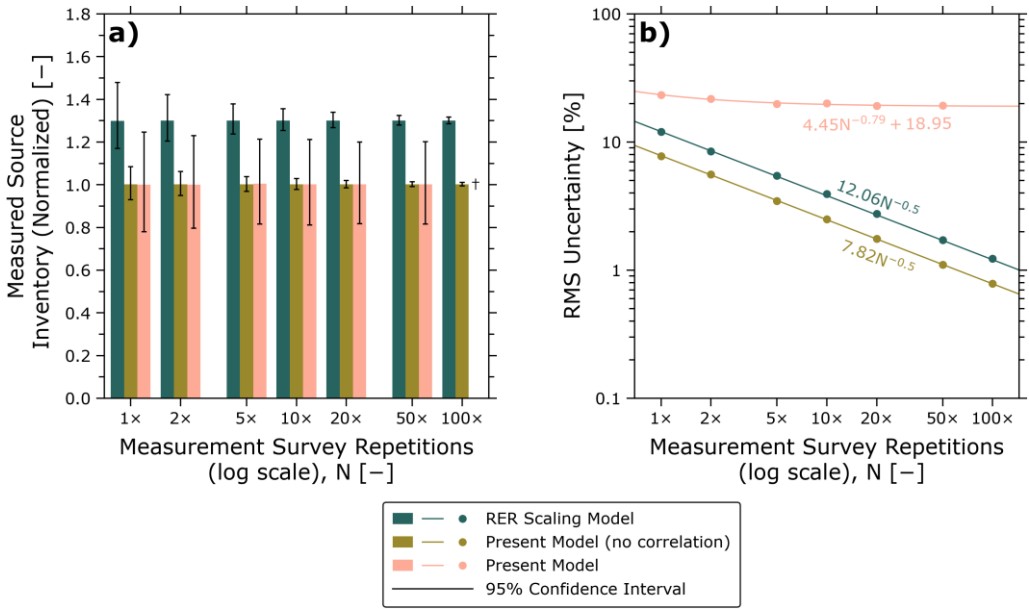

**Figure 5: Effect of the present wind modelling methodology on the measured source inventory for BC in 2021 (Johnson et al., 2023) for remote survey data replicated up to 100× to illustrate the effect of wind speed error correlation. (a) Bar chart with 95% equal tail confidence intervals when using the simple RER scaling model (Conrad et al., 2023b) applied to the present wind speed datasets, the region-averaged wind speed error model (i.e., without autocorrelation), and the full combined model (with autocorrelation) (b) Representative one-sided error (root-mean-square of the upper and lower error bars) for each scenario**
**showing the typical $N^{-0.5}$ decrease in error when correlated errors are not considered and the bounded decrease in error when correlated errors are included via the combined model. †Inventory could not be computed due to computer memory limitations (see Section 4.1).**

Figure 5a shows the three recalculated measured source inventories with 95% confidence intervals, plotted as a function of survey size. There are two key observations. Firstly, we observe that the inventory calculated using the RER scaling

model is approximately 30% larger than when using the combined wind error model. This is a consequence of the former's simplicity, in which the expected value of the ground-truth wind speed is a constant factor multiplied by the NWP model-estimated wind speed. The non-linear conditional means of Figure 2 show that this is a gross over-simplification and, for the present case study at least, results in a sizeable upward bias in wind speed and therefore methane inventory. By contrast, and




as expected, the nominal inventory using the combined model with or without correlation are equivalent (since, for a well-fitting model, correlation has no effect on the conditional mean of the ground-truth wind speed). However, as the measurement survey is artificially repeated, inventory uncertainties when using the RER scaling model or when using the region-averaged wind speed error model alone (i.e., without autocorrelation) both unrealistically shrink to zero. This is best summarized by Figure 5b which plots the RMS uncertainty (i.e., RMS of the upper and lower error bars) as a function of survey repetitions. The figure shows that when correlated errors are ignored, uncertainties decrease per the standard error of the mean ($N^{-0.5}$) as is expected for the inventory calculation, which is essentially a linear function of mean emission factors for various facility/well types. We also observe that the RER scaling model provides larger uncertainties than the region-averaged wind speed error model when autocorrelation is ignored; this is due to the complexity of the latter that captures non-linearity in the relation between ground-truth and NWP model-estimated winds.

Most importantly, Figure 5b shows the dramatic effect of wind speed error correlation. For the nominal inventory without repetition, uncertainties increase by a factor of approximately three when correlation in the wind speed data is accurately modelled. Perhaps more importantly, correlations in wind speed impose a lower bound on uncertainties — more measurements do not bring uncertainties to zero. This is a desired characteristic of the combined model. While, in general, additional measurements tend to improve accuracy in the nominal inventory, because wind speed errors measurements are not generally statistically independent (i.e., they are correlated), central limit theorem does not apply, and the uncertainties should not be expected to decrease to zero ad infinitum with additional measurements.

## 3.4 Temporal variability of the case-study wind speed error model

To assess the robustness of the combined model to variations across different years, we have derived combined models using the same methodology for data during each May–October period of 2021 through 2024 independently. Figure 6a plots the bias (mean error with a positive value indicating overestimation by the NWP model) and standard deviation of the region-averaged wind speed error models (i.e., $\pi^*(u|\tilde{u})$) as a function of NWP model-estimated 10-m wind speed for these periods of time and for the aggregated data from 2021−2024. Encouragingly, there is little variation in both the central tendency and variability across years, with the aggregate result representing a good temporal average. This is true also for the spatial and temporal correlograms in Figures 6b and 6c with perhaps the largest deviations being apparent in the spatial near-field which may be related to the limited number of weather stations and the scattered empirical spatial semivariogram (see Section 3.2 and Figure 4). Overall, the minimal observed deviations in Figures 6b and 6c suggest that the aggregated model provides a robust representation of spatiotemporal correlation for the case study ROI across different years. Leveraging this insight, Appendix A provides six additional combined models derived using the present methodology for the Canadian province of Alberta, the U.S. state of North Dakota, the combined oil and gas region of Saskatchewan and North Dakota (as a surrogate for the Canadian Province of Saskatchewan, which has limited public wind speed measurement data), and the country of Colombia.





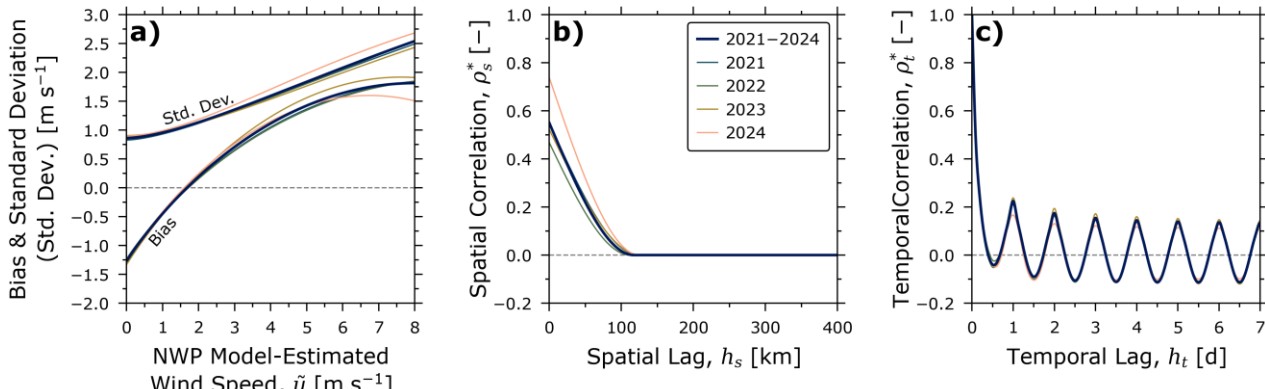

**Figure 6: Comparisons of optimized models using data across four years independently and in aggregate: a) bias and standard deviation of the region-averaged wind speed error model, b) spatial correlogram, and c) temporal correlogram. Results identify modest variability from year-to-year, with the aggregate model representing a good time-average of wind speed error and spatial autocorrelation.**

## 3.5 Comparing performance of NWP models

The derived methodology can be used to assess and compare the performance of different NWP models within a fixed ROI. As a proof-of-concept, two of the six additional analyses described in Appendix A compare the performance of NAM12 and the High-Resolution Rapid Refresh model (HRRR; Dowell et al., 2022; James et al., 2022) in the U.S. state of North Dakota. Figure 7 compares these combined models against the BC case study in the same manner as Figure 6. Firstly, comparing the NAM12 results in BC and ND, we see markedly different results with less overall bias, slower spatial decorrelation, and less pronounced temporal correlation in ND. We suspect that these observations are a result of the very different topographies in BC and ND, with BC being much more rugged and treed and generally exhibiting a lower albedo/surface reflectivity. A higher albedo in ND theoretically reduces surface heating resulting in less diurnal variation in wind speeds; we also observe low temporal correlation in the oil and gas-producing region of the Canadian province of Saskatchewan (see Supplement), which has a similar topography and land cover as ND.





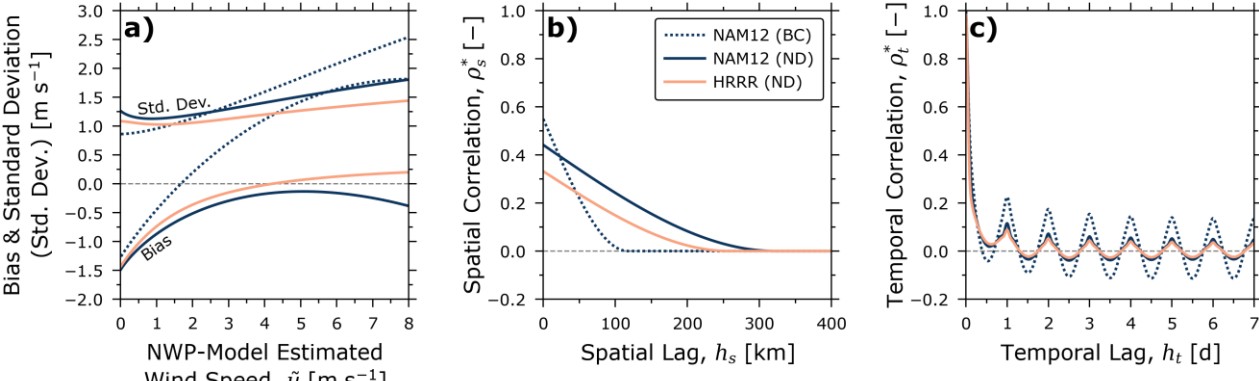

**Figure 7: Comparisons of optimized models for two regions (British Columbia (BC) and North Dakota (ND)) and NWP models (NAM12 and HRRR): a) bias and standard deviation of the region-averaged wind speed error model, b) spatial correlogram, and c) temporal correlogram. Unsurprisingly, performance of the same NWP can be markedly different in different regions.**
**Moreover, the higher-spatial resolution HRRR model provides less bias, variance, and correlation than NAM12 suggesting its usage is preferred for North Dakota.**

The present methodology effectively calibrates NWP model-estimated wind speeds using the available measurement data. However, NWP models do indeed perform differently with respect to bias and dispersion of the region-averaged wind speed error model and spatiotemporal autocorrelation. In ND, surely due to its improved (3-km) resolution and hourly data

assimilation, the HRRR model delivers less bias and dispersion in the region-averaged wind speed error model, faster decorrelation in space, and less overall autocorrelation in both space and time. This demonstrates that when using NWP model-estimated data for large-scale remote surveys and the development of methane inventories – and despite calibration using the same ground-truth dataset – reductions in inventory uncertainties can be achieved by quantifying emission rates using the NWP model that minimizes dispersion in wind speed error and spatiotemporal autocorrelation.

**4  Conclusions**

In this article, we have presented a new method to probabilistically model the errors of NWP model-estimated wind speeds and their autocorrelations in both space and time. The combined model (superimposing the region-averaged wind speed error model and the spatial autocorrelation model) enables random sampling of true wind speeds that are spatiotemporally correlated, which is necessary to properly characterize uncertainties when aggregating remote measurements from large

aerial and satellite surveys into methane inventories. We illustrate the importance of considering spatiotemporal correlation for remote measurement surveys via our case study for the oil and gas-producing region in the northeast of British Columbia, Canada, which demonstrates how bias and precision errors in NWP model-estimated wind speeds can be easily characterized while considering error correlation. The case study model shows notable spatial autocorrelation within a range of approximately 100 km and oscillatory temporal autocorrelation following the diurnal cycle. This former observation

provides guidance for repeated remote measurements in an aerial survey, where, for the case study region, spacing between re-measurements would ideally be greater than two days and phase-shifted by 6 hours to minimize temporal correlation in



the inferred emission rates. We also provide guidance in using the case study models and six additional models for other regions and NWP models (see Appendix A) using the provided code (see Code and data availability).

## 4.1   Limitations and future work

As noted in Section 2.1.2, there is some ambiguity regarding the averaging time of ground-truth measurement wind speeds

for the present case study; while averaging time is generally recommended to be ten minutes, we cannot be certain that this standard is applied for all ground-truth data. As different averaging times may be desired for different remote measurement techniques, the applicability of the derived case-study model might reasonably be questioned. However, on the scale of a measurement, the averaging time would only affect the dispersion in the derived region-averaged wind speed error model, the bias would likely be negligibly affected. If the averaging time of the modelled data were smaller than required by the

measurement technique, then dispersion of the region-averaged wind speed error model would be overestimated and vice versa. Ideally, higher resolution ground-truth wind speed data could enable investigation into the significance of this limitation, but these data are not available to our knowledge.

One drawback of the combined models is that, relative to simpler models that do not consider spatiotemporal autocorrelation of wind speed errors, computation time and memory usage for random sampling within a Monte Carlo

framework (see Section 2.3) may be prohibitively large. Random sampling of the multivariate normal distribution requires a decomposition of the $n \times n$ covariance matrix (e.g., Gentle, 2009); the fastest method to do this using the NumPy software package (Harris et al., 2020), the Cholesky decomposition, is of order $n^3$ (e.g., Watkins, 2002). Thus, as $n$ increases, computational time might become prohibitively large. Moreover, even when the Cholesky decomposition is performed "in-place", memory usage scales quadratically, which may cause out-of-memory conditions on desktop computers. This

occurred as part of the analysis in Section 3.3 (see rightmost data point in Figure 5a), where the simulated survey with 162,600 measurements caused memory overflow. Practically, however, memory overflow issues like this can be solved by leveraging modern high-performance scientific computing resources.

To model spatiotemporal autocorrelation with the Gaussian copula through the semivariogram, we assumed second-order stationarity of the transformed data $\mathbf{z}$ (refer to Section 2.3). In general, we do not expect wind speed errors for any one

weather station and time period to be stationary since, for example, microscale variations in topography would likely cause bias and precision error in the NWP model-estimated wind speed that change with space and time. The same can be said when considering data across regions and time periods in general, owing to numerous factors including the finite availability of measurement data and, more importantly, errors in NWP model-estimated wind speeds that change in time and space. While the region-averaged wind speed error model $\pi(u|\tilde{u})$ considers the average bias and precision errors, it does not

remove the spatiotemporal trend of these from $\mathbf{z}$. As such, the assumption of secondary stationarity is overly optimistic and there is likely some error in the combined models because of non-stationarity in the data. We also (see Eq. (15)) model spatial autocorrelation as isotropic. Given the non-uniformity of terrain in general but especially in the case study ROI, we





would expect spatial autocorrelation to be anisotropic; however, to maintain the tractability of our modelling approach, we ignore this and accept corresponding inaccuracy in the model.

Our case study and additional analyses derive models for wind speeds at 10-m above ground level. The modelling approach, however, is agnostic to the specific height above ground level and model fitting could be performed using any height for ground-truth and NWP model-estimated winds; they could indeed even be different. The caveat is that additional bias and precisions errors will be engendered if scaling any wind speeds using an assumed wind speed profile.

One opportunity to improve our methodology is to expand the candidate models for the marginal wind speed error distributions (see Section 2.2.2). Referring to Figure 3, the performance of the case study model reduces as the NWP model-estimated wind speeds go to zero. This is partly due to the reduction of available data in this domain – i.e., model fitting is naturally biased by the amount of available data – but may also be a consequence of the candidate functions for $\boldsymbol{\theta}(\tilde{u})$ (constant, linear, or offset power-law). Theoretically, the model might be improved by considering additional candidate functions of different forms.

## Appendix A: Additional analyses

To test the robustness of the present methodology for different regions of interest (ROIs), glean insight into the spatial variability of wind speed error and autocorrelation, and provide combined models for methane inventory development in various oil and gas-producing regions, we have developed six additional combined models. These include combined models for NAM12 in three additional ROIs using weather station data for the oil and gas-producing regions of the Canadian provinces of Alberta (AB) and Saskatchewan (SK) and the U.S. state of North Dakota (ND). We also develop an additional combined model for Saskatchewan with ground-truth data augmented by measurement data in North Dakota, which has a similar topography and oil and gas sector; we denote this analysis as "SK+ND". To support parallel work of the authors' in developing a methane emissions inventory, we also present a combined model for the primary oil and gas-producing region of Colombia (COLO) using the ERA5-Land reanalysis (Muñoz Sabater, 2019) from the European Centre for Medium-Range Weather Forecasts (ECMWF). Finally, to illustrate the effectiveness of the present method for comparing NWP model performance, we also develop a combined model for ND using the high-resolution rapid refresh (HRRR) forecast system. Additional figures corresponding to Figure 2, 3, and 4 for each of the additional analyses are available in the supplement.

Ground-truth measurement data supporting the additional analyses were sourced from the ECCC network noted in the manuscript augmented by data from three weather station networks: the Ambient Air Quality network of the Government of Alberta (AB-AAQ), the North Dakota Agricultural Weather Network (NDAWN), and Colombia's Instituto de Hidrología, Meteorología y Estudios Ambientales (IDEAM). Table A1 summarizes the ground-truth data for these analyses in the same manner as Table 1.



**Table A1: Ground-truth data sources providing scalar wind speeds for comparison with NWP model-estimated forecasted wind speeds and development of combined models for additional jurisdictions.**

| Relevant Jurisdiction | Authority/Network | Acronym | Count | Notes |
|---|---|---|---|---|
| Alberta | Alberta Ambient Air Quality Data | AB-AAQ | 129 | — |
| | Environment and Climate Change Canada/Meteorological Service of Canada's Surface Weather Stations | ECCC | 31 | a,b |
| Saskatchewan | | | 19 | a,c |
| | North Dakota Agricultural Weather Network | NDAWN | 89 | d |
| North Dakota | | | | — |
| Colombia | Instituto de Hidrología, Meteorología y Estudios Ambientales | IDEAM | 253 | e |

ᵃ The anemometer is *usually* located at 10 m above ground. Measurements represent the one-, two-, or 10-minute period preceding the top of the hour (Technical Documentation: Historical Hourly Climate Station Data, 2025).
ᵇ Excludes 6 weather stations within 10 km of known NAM data source(s).
ᶜ Excludes 8 weather stations within 10 km of known NAM data source(s).
ᵈ One additional analysis of Saskatchewan (SK+ND) augments ECCC data in Saskatchewan with NDAWN data.
ᵉ Model developed using data over the entire year for 2021–2024 for Colombia, where snow does not preclude aerial survey.

**Code and data availability**

A python code summarizing model equations and enabling visualization and Monte Carlo sampling (per Section 2.3) of the

developed models is provided as part of this article's supplement located at https://doi.org/10.5683/SP3/PMLX4X.

NAM12 data are publicly available via https server at https://www.ncei.noaa.gov/data/north-american-mesoscale-model/access/forecast/ and on Amazon Web Services (2021-09-16 and later) at https://registry.opendata.aws/noaa-nam/. HRRR data are publicly available on Amazon Web Services (2014-07-30 and later) at https://registry.opendata.aws/noaa-hrrr-pds/. ERA5-Land data are publicly available through ECMWF's Climate Data Store at

https://cds.climate.copernicus.eu/datasets/reanalysis-era5-land. Climate/weather station data published by Environment and Climate Change Canada, the Government of British Columbia, the Government of Alberta, the North Dakota Agricultural Weather Network, and Colombia's Instituto de Hidrología, Meteorología y Estudios Ambientales are publicly available through online portals, anonymous FTP access, and/or https server:

- ECCC: https://dd.weather.gc.ca/climate/observations/hourly/csv/BC

- BC-ENV (Contains information licensed under the Open Government Licence – British Columbia):
       ftp://ftp.env.gov.bc.ca/pub/outgoing/AIR/AnnualSummary/ (verified) and
       ftp://ftp.env.gov.bc.ca/pub/outgoing/AIR/Hourly_Raw_Air_Data/Year_to_Date/ (unverified)
- BC-FLNRO (Copyright © 2025, Province of British Columbia):
       ftp://ftp.for.gov.bc.ca/HPR/external/!publish/BCWS_DATA_MART and

https://www.for.gov.bc.ca/ftp/HPR/external/!publish/BCWS_DATA_MART/
- AB-AAQ: https://datamanagementplatform.alberta.ca/Ambient
- NDAWN: https://ndawn.ndsu.nodak.edu





- IDEAM: https://www.datos.gov.co/Ambiente-y-Desarrollo-Sostenible/Velocidad-del-Viento/sgfv-3yp8

**Author contributions**

BC conceptualized the research and developed the methodology. MJ was responsible for funding acquisition, project administration, provision of resources, and supervision. BC curated the data, performed the formal analysis and investigation, and developed the software. Both authors produced the original manuscript and reviewed and edited the manuscript throughout the publication process.

**Competing interests**

The authors declare that they have no conflict of interest.

**Acknowledgements**

This work was supported by the United Nations Environment Programme (UNEP) under the framework of UNEP's International Methane Emissions Observatory (IMEO, Grant DTIE22- EN4582), the British Columbia Government through the Ministry of Environment and Climate Change Strategy (Grant No. TP23CASG0011MY), the Natural Sciences and Engineering Research Council of Canada (NSERC, Grant Nos. ALLRP 590391-23 and RGPIN-2024-06485), and Natural Resources Canada (NRCan, Grant No. EIP-22-002).

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
