# Peer review of "Accounting for spatiotemporally correlated errors in wind speed for remote surveys of methane emissions"

_EGUsphere, 2025_

## Author Comment (AC1)

**Journal:** Atmospheric Measurement Techniques
**Manuscript ID:** 2025-3924
**Title:** Accounting for spatiotemporally correlated errors in wind speed for remote surveys of methane emissions
**Authors:** Bradley M. Conrad and Matthew R. Johnson

**Point-by-point Responses to Reviewer Comments**

**Reviewer 1**

*In their work Conrad and Johnson handle the importance of error correlation in wind speed data when deriving Methane emissions from measured concentrations. An algorithm to quantify spatiotemporal auto-correlation is described, giving guidelines on how to best perform measurement campaigns in certain regions of interest. While this study mainly focuses on the methane emission example, the core method is applicable for any method that relies on model wind data, further underscoring the scientific significance of this work. Overall, the presentation quality of this study is excellent. In the following some minor revisions and technical corrections are suggested that mainly focus on improving the understand-ability of the study.*

We thank the reviewer for their helpful feedback and technical comments, for noting the significance and applicability of the methodology to researchers in other fields, and for commending the presentation quality of our manuscript.

*Minor revisions:*

*As a reader who is not proficient on measurement statistic algorithms, section 2 would largely benefit from a more tangible explanation approach using less mathematical detail and more graphics that explain the used methods. It would be very helpful to have a clear recipe of what is needed to apply the described algorithm (e.g. NWP data on a fine grid with high temporal resolution and statistically independent station data).*

The methods section leverages the mathematical detail necessary to clearly describe and explain the methodology. To aid the reader we have organized the text to begin by describing our case study (Section 2.1.1), detailing the ground-truth (Section 2.1.2) and NWP (Section 2.1.3) data being analyzed, and how such data are "pre-processed" (Section 2.1.4). We continue by outlining the high-level approach to considering spatial autocorrelation (Section 2.2.1) and provide detailed descriptions of the modelling steps (Sections 2.2.2 and 2.2.3) before finally noting how the model can be used in practice (included step-by-step pseudocode in Section 2.3).

In addition to noting the requirement for NWP and weather station data as a limitation of our method (see our response to the reviewer's last minor comment), we have revised the introduction to also explicitly note this data requirement (additions in bold):

*In this manuscript, we detail a methodology to probabilistically model the true wind speed  **in** an arbitrary  **region** and **during an arbitrary** time **period** from gridded, discrete-time NWP model estimates **and statistically independent weather station data**.*

Finally, as now specifically noted in the revised text, code used in our analysis can be shared upon request.

*While the authors give a clear explanation on the importance of error calculation for the resulting Methane emission estimate, an estimate on how the described method compares to other sources of uncertainty could provide more insight in said importance. Examples for other sources of uncertainty in emission calculation: Injection height and resulting usage of the wind field (speed and direction), missing "measured" wind data in the atmosphere above ground level, uncertainty of the measured concentrations.*

The reviewer is right to assert that wind speed uncertainty is only one contributor to the overall "uncertainty budget" of methane emissions quantification. However, other sources of uncertainty are dependent on the measurement technique being employed and the (potentially proprietary) quantification algorithms; a generalized comparison of wind speed uncertainty against other sources of uncertainty would be challenging to fairly create and is out-of-scope for this work. This present method is intended to provide robust uncertainties for the wind speed uncertainty contribution alone.

*Section 3.5 nicely shows the effect of model resolution on the wind speed error model. However, the aforementioned comparison to other sources of uncertainties could provide information on how the importance of the described model changes for different spatial or temporal resolutions of the NWP data set. This would also help the reader to understand the importance of the wind speed error model.*

Please see our previous response regard the challenge of generalized uncertainty comparison across different measurement techniques and quantification algorithms.

*A methane emission calculation comparison between the following three approaches would further the understanding of the importance of using a wind error model: detailed handling of error calculation (main topic of this study), a simple approach to error handling (probably similar to RER approach described in the study) and the approach of neglecting wind error.*

Section 3.3 of the submitted manuscript provides a detailed comparison of the simple error handling approach (the RER approach referred to by the reviewer) and the new methodology without and with consideration of error correlation. We specifically chose not to compare against the reviewer's third recommendation (neglecting wind error) as uncertainties in this case are not generalizable; they are highly dependent on the measurement technique, survey size, and emissions profile in the region of interest. Moreover, when excluding (correlated) wind speed errors, aggregated random errors in an inventory application quickly and unrealistically average toward zero, especially for high-precision active sensors like Bridger Photonics' Gas-Mapping LiDAR, which was used in our case study.

*The work motivates why a model of the wind speed error is important and how to best apply the gained knowledge, e.g. in planning of measurement campaigns. However, I'd like to see at least a small focus on how to handle imperfect conditions: What do I do if I don't have an independent measurement data set in addition to a NWP using data assimilation? Is it possible to generalize some of the found features? Maybe using parameters like surface roughness, main wind direction and topography?*

This is a terrific point. While there would always be some available NWP model (there are models with global coverage), there are certainly regions where wind speed measurement data are not publicly

available. Characterizing specific NWP models as a function of the confluence of prevailing winds, topography, surface roughness, etc. is beyond the scope of this work; however, in these situations we would suggest seeking representative ground-truth wind speed measurements from a *similar and nearby* region, if possible. In this scenario, we would of course expect potential bias in the model of error and its correlation, which would unfortunately be challenging to robustly quantify. We have added the following paragraph to the limitations section to address this.

> *The methodology we have outlined requires ground-truth wind speed measurement data in the specific region of interest. We expect that such data do not exist or are not publicly available in some regions. In such a case, we would recommend that representative wind data be sought from a nearby region with similar topography and prevailing winds, if possible, while recognizing that there would be some unquantifiable bias in the model of wind speed error and its autocorrelation.*

***Technical corrections/suggestions:***

*Page 1 Lines 20—24: Instead of providing the finding of how to best perform measurements w.r.t. correlation, the estimate on how large the emission uncertainty increases if neglecting wind speed error correlation would in my opinion be beneficial for this study.*

We agree that this is a key result of this manuscript. But, estimating the change in emissions inventory uncertainty when considering wind speed error correlation is challenging to generalize as it theoretically depends on the region of interest, the time of year, the measurement technique (i.e., how wind affects quantification), and the size of the survey. We have added the following text to the abstract to refer to this key result for our case study, without providing an explicit magnitude:

> *We observe in our case study region that correlation in wind speed errors can starkly increase overall uncertainties in emissions inventories, especially for large surveys.*

*Page 2 Lines 3—24: I'm missing a step in between describing the common challenge and why/how much the correlation of "wind speeds (and hence their uncertainties)" affects the emission. Maybe the authors could give an example emission calculation from given methane enhancements. This could help to better explain where in that calculation, correlation of measurements and underestimation of the wind speed error affect the derived emission.*

At lines 3−16 of the original text, we present the importance of wind speed in the calculation of emissions and how NWP models can be a key contributor to uncertainties in emissions. At lines 17−24 of the original text, we then discuss how *correlation* in wind speed (and, therefore, emission rate) errors cannot be ignored when "aggregating sources to produce an inventory". We believe that the potential source of confusion surrounds what happens if correlation is ignored in large surveys. We have revised the text to now explicitly note the effect of central limit theorem when aggregating with uncorrelated errors (additions in bold):

> *Neglecting this autocorrelation when aggregating sources to produce an inventory will**, through central limit theorem,** artifically reduce the contribution of wind speed precision error  **and hence** the overall uncertainty of the inventory.*

*Page 2 Line 11: I had difficulties finding the work from Branson et al., 2021. The other example for an aerial measurement approach (Thorpe et al., 2021) describes methane emission estimates using a LiDAR*

*technique, while LiDAR is separately mentioned in the second half of the sentence. The currently sentence suggests that these two methods are different, but the references point to the same measurement technique.*

Thank you for identifying this. We intended to reference a different work by (Thorpe et al., 2023) and have updated the text accordingly. We have also removed the reference to Branson et al., which is a white paper by Kairos Aerospace (now Insight M) that we can also no longer source, in favour of a reference by (Duren et al., 2019) that explicitly describes methane plume quantification with airborne imagers.

*Page 17 Line 11: […] of their semivariogram (left axis) and the their correlogram […] - remove the "the" after "and"*

Revised, thank you.

*Page 18 Line 2: […] the spatial correlogram is trivially calculated by […] - remove the "trivially" after "is"*

Revised as recommended.

*Page 19 Lines 7/8: […] At large lags, temporal correlations, representing bias over the diurnal cycle, oscillate with an amplitude of approximately 0.13. […] – is there a physical reason for this diurnal bias? Is it connected to sub-model-scale meteorology?*

We suspect that this there are some physical process(es) that are simply not captured by the NWP models. We have revised the text to note this (additions in bold):

> *... representing bias over the diurnal cycle**, presumably due to temporally dependent physical process(es) not captured by the NWP Model***, *...*

**References**

Duren, R. M., Thorpe, A. K., Foster, K. T., Rafiq, T., Hopkins, F. M., Yadav, V., Bue, B. D., Thompson, D. R., Conley, S., Colombi, N. K., Frankenberg, C., McCubbin, I. B., Eastwood, M. L., Falk, M., Herner, J. D., Croes, B. E., Green, R. O. and Miller, C. E.: California's methane super-emitters, Nature, 575(7781), 180–184, doi:10.1038/s41586-019-1720-3, 2019.

Thorpe, A. K., Kort, E. A., Cusworth, D. H., Ayasse, A. K., Bue, B. D., Yadav, V., Thompson, D. R., Frankenberg, C., Herner, J., Falk, M., Green, R. O., Miller, C. E. and Duren, R. M.: Methane emissions decline from reduced oil, natural gas, and refinery production during COVID-19, Environ. Res. Commun., 5, 021006, doi:10.1088/2515-7620/acb5e5, 2023.

---

## Author Comment (AC3)

**Journal:** Atmospheric Measurement Techniques
**Manuscript ID:** 2025-3924
**Title:** Accounting for spatiotemporally correlated errors in wind speed for remote surveys of methane emissions
**Authors:** Bradley M. Conrad and Matthew R. Johnson

**Point-by-point Responses to Reviewer Comments**

**Reviewer 2**

*In this manuscript, Conrad and Johnson discuss the importance of wind speed uncertainties when estimating methane emissions using remote sensing observations. The study is strongly focused on theory and technical elements of model development. They demonstrate the applicability of a combined model for wind uncertainties for a testcase in Canada. They also suggest a new experimental design to reduce temporal autocorrelation effects. The topic is of great importance as this source of uncertainty is usually neglected in remote sensing studies. The text is well written und the manuscript has a clear and easy to follow structure. However, given the focus of the manuscript on model design it might be better suited for another EGU journal, namely, Geoscientific Model Development. Nevertheless, the study also overlaps with the scope of AMT.*

We thank the reviewer for their positive comments.

***There are only some minor comments that should be addressed before publication.***

*Page 1, line 27: Would you consider the level of uncertainty similar for aircraft and satellite studies or should they be considered differently? Especially the fact that airborne surveys often have on-board wind data and are not restricted to clear-sky day bias could suggest that they might not experience the same limitations.*

Uncertainty in the quantification of methane emissions is highly dependent on the measurement technique. For aircraft- and satellite-mounted imagery- and LiDAR-based techniques, uncertainties are driven by spatial resolution of the imagery, the precision of inferred methane concentrations/enhancements, and, as we focus on in this work, the estimation of a representative wind speed that propagates the emission. Although comparing uncertainties between these various techniques is beyond the scope of this work, the contribution of wind speed error is expected to be similar.

Indeed, direct measurement of wind speed could be highly advantageous in reducing uncertainties in estimated emissions. Direct measurement could be performed with ground-based on-site anemometer(s) or advanced remote techniques from the aircraft, such as Thorpe et al.'s (2021) airborne doppler wind LiDAR.

*Page 2, line 17 [Page 4, line 16]: Why is the analysis limited to May to October? Satellites are observing and reporting observations in all seasons. Are you confident there is no seasonal bias in the NWP performance?*

In our case study's region of interest (northeastern British Columbia, Canada), snow-cover generally precludes accurate measurement or, at a minimum renders measurement challenging, during approximately November to April, inclusive. As such, for the case study, we have constrained our analysis period from May to October.

Seasonal bias is certainly a possibility if not a likelihood and we do not suggest that it is absent. To make this clear, we have made two revisions. In Section 3.1, we identify that this time period is chosen the "typically non-snowy months in this region". We also now note in our revised limitations and future work section that:

> *Our case study analysis was performed during the typical non-snowy period in Northeastern British Columbia, May to October, since the presence of snow may preclude remote measurements from air and space. The presented methodology is agnostic to the time period of interest and can be applied to data from any time period as long as there are sufficient data available. Indeed, as indicated in Table A1, we have applied our method to evaluate the ERA5-Land reanalysis product over the entire calendar year in the primary oil and gas-producing region of Colombia.*

Please also note that this methodology is agnostic to the period of interest (Table A1 shows that we execute this analysis for the whole calendar year in Colombia) and can indeed be employed to objectively assess seasonality of wind speed error.

*Page 7, line 14-15: The two references cited here: Sklar 1959 and Nelsen 2006 are not easily accessible or behind a paywall. So, please provide more details on Copulas here or provide additional references that discuss Copulas in more detail.*

These citations refer to books. The work by Sklar (1959), which first presented Copulas, is available through the HAL open archive. The work by Nelsen (2006) is the seminal textbook on the topic with over 20,000 citations but it does not seem to be freely available online.

*Page 9, line 12: More details on DECLUS would be helpful here.*

We have revised the text to read (additions in bold):

> *Data were weighted ... using the "DECLUS" cell declustering algorithm (Deutsch and Journel, 1997)**, which weights station data by the inverse of station count on a regular but randomly perturbed two-dimensional grid**.*

*Page 9 line 19: If there is a myriad of literature, why do you only provide a single reference, which is, again, behind a paywall.*

Cressie (1993) is a seminal textbook on the topic of spatial statistics, which we present as an example of the literature in the field. We have revised the text to note this (additions in bold):

> *... autocorrelated geostatistical data**, including the seminal work of** (Cressie, 1993).*

*Page 12: Section 2.3. highlights that this study is really about the model itself and maybe better suited for Geoscientific Model Development. Nevertheless, it is a good example for more detailed analysis of correlated uncertainties affecting many applications.*

We appreciate the reviewer's feedback.

*Page 19, line 24: The point about sun-synchronous satellites is crucial and it might be good to highlight that nearly all current satellites used for methane emission monitoring are sun-synchronous.*

This is a great point, which we have adopted in the text (additions in bold):

*Of course, this is not possible for observation by sun-synchronous satellites**, which includes most methane-detecting satellite instruments (Jacob et al., 2022),** but could be accommodated by careful planning of aerial surveys.*

**References**

Cressie, N. A. C.: Statistics for Spatial Data, Revised., John Wiley & Sons, Inc., New York, NY., 1993.

Deutsch, C. V. and Journel, A. G.: GSLIB: Geostatistical Software Library and User's Guide, 2nd ed., Oxford University Press, New York, NY. [online] Available from: http://claytonvdeutsch.com/wp-content/uploads/2019/03/GSLIB-Book-Second-Edition.pdf, 1997.

Jacob, D. J., Varon, D. J., Cusworth, D. H., Dennison, P. E., Frankenberg, C., Gautam, R., Guanter, L., Kelley, J., McKeever, J., Ott, L. E., Poulter, B., Qu, Z., Thorpe, A. K., Worden, J. R. and Duren, R. M.: Quantifying methane emissions from the global scale down to point sources using satellite observations of atmospheric methane, Atmos. Chem. Phys., 22(14), 9617–9646, doi:10.5194/acp-22-9617-2022, 2022.

Nelsen, R. B.: An Introduction to Copulas, Springer Science+Business Media, Inc., New York, NY., 2006.

Sklar, A.: Fonctions de répartition à n dimensions et leurs marges, Publ. l'Institut Stat. l'Université Paris, 8, 229–231, 1959.

Thorpe, A. K., O'Handley, C., Emmitt, G. D., DeCola, P. L., Hopkins, F. M., Yadav, V., Guha, A., Newman, S., Herner, J. D., Falk, M. and Duren, R. M.: Improved methane emission estimates using AVIRIS-NG and an Airborne Doppler Wind Lidar, Remote Sens. Environ., 266(August), 112681, doi:10.1016/j.rse.2021.112681, 2021.

---

## Author Comment (AC4)

**Journal:** Atmospheric Measurement Techniques
**Manuscript ID:** 2025-3924
**Title:** Accounting for spatiotemporally correlated errors in wind speed for remote surveys of methane emissions
**Authors:** Bradley M. Conrad and Matthew R. Johnson

**Point-by-point Responses to Reviewer Comments**

**Reviewer 3**

*This manuscript provides a rigorous probabilistic modeling study of numerical weather prediction (NWP) 10-m winds' error relative to independent ground-based wind speed measurements. A region-average wind error model is fitted as a Weibull distribution. Then, the errors are modeled marginally in space and time and combined using Gaussian coupla and spatitemoral semivariograms. It seems technically sound and well written, with the following points to consider for further improvements.*

> We thank the reviewer for their detailed review of this manuscript and for identifying the rigour and technical quality of this work.

*A main issue is that the connection between the core analysis/results of this work and the scope of AMT (specifically remote sensing of methane point source emissions) is relatively weak. It is really section 3.5 only. It may be helpful to provide more context on how this work fits in the methane quantification pipeline. Specifically, the 10-m wind that this study tries to model, as the ground truth, is measurement at (usually) 10 min interval. The 10-m wind->effective wind->methane emission pipeline is calibrated all in an LES model in Varon 2018. To what extent is the "ground-truth" wind speed relevant, if the wind-emission relationship is calibrated by a model? This study tackles the error from NWP to ground-measured 10 m wind, and it would be nice to have some discussion on the mapping from measured wind to effective wind and then to emissions. It almost makes me feel that targeting measured wind is a detour, and one should map NWP wind to LES wind, or whatever wind that calibrates the wind-emission relationship.*

> We believe that this manuscript is well within the scope of AMT, specifically (from AMT's stated aims and scope) the "validation … of techniques of data processing" relevant to the retrieval, here quantification, of atmospheric gases.

> We explicitly state in our introduction where wind quantification exists in the methane plume quantification pipeline. We would also argue that the ground-truth wind field, which is what has propagated an observed plume, is the exact wind data required to quantify methane flux through a control surface.

> Varon et al. (2018) – which importantly is just one approach to bring wind speed information into a quantification pipeline as we note in our introduction – developed "calibration" functions that estimate an "effective wind speed" from a ground-truth wind speed. In their simulation work, the ground-truth wind speed was necessarily modelled using LES but, in practice, absent local measurements, this ground-truth wind speed would be estimated using an NWP model. Thus – even for this unique calibration approach that, to our knowledge, is not commonly used by others – evaluating the performance of NWP models vs. ground-truths is critical.

*Fittings of the regional wind error model and the semivariograms are important for this work, so it is recommended to include details on how those fitting algorithms, specially the numerous constraints in fitting parameters, are implemented.*

As noted by the reviewer below (see text following equations 17, 18, and 19 in the manuscript), we do explicitly note the constraints on fitted parameters. For optimization, we [use] MATLAB's *fmincon (constrained minimization) function* for the maximum likelihood estimation. We have revised section 2.2.2 to note this.

***Technical comments:***

*Eq. 8: please double check as $\pi$ seems to be reserved for a PDF, and dividing a PDF by u~ is unlikely to give another PDF.*

The equation is correct as written. This is a change of variables to the distribution $\pi_{cand}$ and the $1/\tilde{u}$ multiplier is the derivative of RER, necessary to ensure the law of total probability for the transformed distribution.

*Page 9, lines 10-11: it is recommended to provide the formula of AIC as the objective function.*

The AIC (Akaike Information Criterion) is a standard tool to evaluate relative model performance in a Bayesian setting and is essentially a parameter count-corrected negative log-likelihood. Rather than introduce the formulation for AIC, we have revised the text to include this description.

*Page 10, line 1: should that be the "... the variance of the difference of the zi at these positions"?*

Yes, revised as requested.

*Page 11, lines 9-10: double check the location and span of bins. Should that be 0-50 km and 25-75 km?*

Yes, thank you for catching this. Revised.

*Page 11, lines 24-25: H seems to be reserved for joint CDF. Consider another font/symbol for Heavyside step function.*

Thank you for catching this. We have revised the equation to use a $R$ for the Heaviside step function.

*Page 11, lines 26: please confirm what $\gamma_{k'}(b;b)$ is, and why it should be 0.95.*

Since the monotonic semivariogram models employed only approach a value of unity asymptotically, their *range*, which defines a spatial limit of correlation must be defined using some threshold. By convention, this is typically taken as 0.95, which we employ here. Noting that $a_k$ and $b$ are always coupled, any value in (0, 1) could theoretically be chosen without affecting the underlying function; the fixed constants $a_k$ and optimized range, $b$, would simply be different. We have added text to note that 0.95 is chosen by convention.

*Page 12, lines 1-2: fixing b3 to 0 contradicts the constraint (b3>b2>b1>=0) in the previous page. Please elaborate.*

Thank you for catching this. We have revised to note that when each model is considered independently, $b_3$ is not required.

*Page 12, lines 17-18: close the parenthesis.*

Revised, thank you.

*Page 21, line 5: should the RMS uncertainty be read as the "distance" of the upper and lower error bars?*

As noted in the text, the RMS uncertainty is the root-mean-square of the upper and lower error bar lengths. We choose this parameter to collapse the asymmetric uncertainties into a single parameter.

*Section 4.1: it reads a bit strange to have such a dominant subsection in conclusion section. Consider making it a dedicated section preceding the conclusion.*

We have revised the structure as recommended to place this limitations subsection as section 4 preceding the conclusion.

**References**

Varon, D. J., Jacob, D. J., McKeever, J., Jervis, D., Durak, B. O. A., Xia, Y. and Huang, Y.: Quantifying methane point sources from fine-scale satellite observations of atmospheric methane plumes, Atmos. Meas. Tech., 11(10), 5673–5686, doi:10.5194/amt-11-5673-2018, 2018.